# Can the Synergy of Digitalization and Servitization Boost Carbon-Related Manufacturing Productivity? Evidence from China's Provincial Panel Data

Gang Li *, Yanan Chen and Yan Cheng

School of Economics Management, Xi'an University of Posts & Telecommunications, Xi'an 710061, China;
chenyanan@stu.xupt.edu.cn (Y.C.); 553413420@stu.xupt.edu.cn (Y.C.)
* Correspondence: cligan@xupt.edu.cn

**Abstract:** With the goal of carbon peaking and neutrality, carbon productivity has become a means of sustainability in manufacturing, and the impact of the synergy of digitalization and servitization (DSS) on carbon productivity (CP) deserves in-depth study. Based on data with respect to manufacturing in 30 provinces in China from 2013 to 2020, a coupled coordination degree model is used to calculate the degree of manufacturing coordination. A regression effect model is used to explore the intrinsic mechanism of the impact of DSS on CP. The main results show the following: (1) The DSS in manufacturing positively contributes to enhancing CP, and there are non-linear features in both. (2) Technological innovation can contribute to the impact of DSS on CP, as does industry structure, and there is a mediating effect between the two. (3) When economic growth is used as the threshold, DSS and CP reflect a positive "U" relationship. Based on the above findings, policy recommendations are made to promote the sustainable development of manufacturing.

**Keywords:** synergy of digitalization and servitization; coupling coordination degree; positive "U" relationship; carbon productivity; manufacturing



## 1. Introduction

Serious environmental degradation has accompanied economic development [1]. Economic growth has increased the emission of greenhouse gases [2], of which $CO_2$ is representative, exacerbating harm to the ecosystem. Environmental protection has also become a widespread concern in the world. China has become a large $CO_2$-emitting country due to its rapid economic development and industrialization [3]. China proposes carbon peaking and carbon neutrality targets to realize global environmental protection [4]. In order to steadily achieve this goal, manufacturing in China needs to realize green and sustainable development [5]. Carbon productivity improvement can be used as a sustainable development indicator to coordinate economic growth and reduce $CO_2$ emissions. It is a sustainable development path for the green development of manufacturing in China [6].

Carbon productivity is mainly concerned with the economic efficiency of carbon emissions and has a reciprocal relationship with carbon emissions. Its essence lies in achieving the maximum economic output while minimizing carbon emissions [7]. The synergy of digitalization and servitization can be a viable path to increasing carbon productivity. Digitalization has the potential to enhance the carbon productivity of manufacturing by improving its R&D intensity and efficiency [8]. Servitization, the process by which manufacturing firms integrate services into their offerings, can indeed contribute to increasing carbon productivity by improving their competitive advantage [9]. Enterprises harness the synergies between manufacturing and productive services to broaden and deepen their impact in the industry, thereby fostering sustainable development [10]. This includes optimizing industrial layouts through industrial synergies to increase carbon productivity [11],

and it also includes promoting high-quality manufacturing through digitalization and servitization synergies and increasing carbon productivity [12].

In an analysis of the influence path of digitalization and servitization on carbon productivity, both industrial structure and technological innovation have important transmission roles. Technological innovation can delay the emergence of the servitization paradox and promote the servitization of manufacturing [13]. Similarly, technological innovation positively regulates the influence of manufacturing digitalization on carbon emissions [14] and improves the carbon productivity of manufacturing. The digitalization and servitization of manufacturing can promote a low-carbon economy and improve carbon productivity by optimizing industrial structures [15,16]. As a result, technological innovation and industrial structure can be an effective pathway for the synergy of digitalization and servitization to enhance carbon productivity. Drawing from the environmental Kuznets curve theory, the investigation into economic growth and carbon emissions reveals that the relationship between them is mostly nonlinear. Yu studied economic growth as a threshold variable and found the threshold effect of manufacturing on carbon emissions [17].

Most research involving carbon productivity focuses on the enhancement effect of digitalization [8]. Digitalization improves carbon productivity by fostering technological innovation and transforming industrial structures [14]. There is a scarcity of research focusing on the enhancing effect of servitization on carbon productivity [9]. Research on the specific path of enhancing carbon productivity is unclear. Current research mostly focuses on a single dimension of digitalization or servitization to verify the impact on carbon productivity, and there is a lack of relevant research on both dimensions. We aim to fill the research gap by exploring the impact of the synergy between digitalization and servitization on carbon productivity.

To sum up, this paper reviews the theoretical mechanism of the synergy of digitalization and servitization (abbreviated as DSS) affecting carbon productivity (CP) in manufacturing and explores the following aspects: First, based on the coupled coordination degree model, the synergy level of digitalization and servitization in manufacturing is measured across provinces in China, and the development degree of DSS in each province is clarified. Second, the regression effect model is used to verify the influence of manufacturing DSS on CP. Third, in the intrinsic mechanism of the impact of DSS on CP, the mediating effects of technological innovation and industrial structure are further investigated, and we verify the threshold effect of economic growth in the effect of DSS on CP. This will provide theoretical reference and advice for eliminating the development gap of DSS among provinces and enhancing the CP of manufacturing.

The first part provides a literature review, which discusses the literature related to DSS and carbon productivity. The second part measures the degree of manufacturing DSS and evaluates the developmental status of manufacturing DSS in the provinces of China. The third part uses a dual fixed-effect model to explore the internal mechanism of the impact of DSS on CP in manufacturing. Finally, we offer pertinent policy recommendations to furnish manufacturing with a theoretical framework and guidance for advancing carbon productivity.

## 2. Literature Review and Hypothesis

### 2.1. Manufacturing DSS and Carbon Productivity

The digitalization and servitization of manufacturing exert a beneficial influence on the expansion of carbon productivity, and the relationship between digitalization, servitization, and carbon productivity does not comprise simple linear relationships [18,19]. Through the value-added and substitution effects of digitalization, manufacturing can improve operation efficiency, reduce unnecessary energy consumption, and improve carbon productivity [20]. Heo and others [21] believe that digitalization can directly promote technological innovation, and technological innovation and industrial structure positively contribute to a reduction in carbon emissions and the enhancement of carbon productivity [18]. On the one hand, the development of digitalization needs the support of technology, infrastructure,

and other aspects. The development of digitalization, while increasing economic growth, also increases carbon emissions. The contribution to carbon productivity in manufacturing is therefore not significant [22,23]. On the other hand, digitalization makes rational use of resources in manufacturing by means of digital technology, which reduces energy consumption and improves energy utilization efficiency [24,25]. This promotes industrial economic growth and improves carbon productivity [26]. Similarly, manufacturing promotes the development of servitization and carbon productivity based on its own technological and industrial advantages [27]. Servitization can curtail energy consumption and diminish carbon emissions within the manufacturing sector by increasing the input of service elements [28]. Technological innovation, energy structure optimization, and other means are utilized to curtail energy consumption in manufacturing, enhance energy efficiency, and increase carbon productivity [29]. However, some scholars have pointed out that the technology required for the servitization of manufacturing to develop can accelerate resource consumption, creating the servitization paradox [30]. This makes servitization and carbon productivity reflect a non-linear relationship.

Current studies on carbon productivity found that there are positive "U" [31], positive "N" [32], and other characteristics of carbon productivity. Xu and others [31] confirmed that the interplay between manufacturing and producer services, along with carbon productivity, exhibits "U" characteristics. Only a high level of synergies can promote carbon productivity. Song and Han [32] found that environmental regulation has an "N" relationship with carbon productivity across time and geography. Zhang [33] proposed that there exists a positive "U" feature between economic growth and carbon productivity, but the conclusion has not been further empirically tested.

Based on the above theories, it becomes evident that digitalization and servitization exert a positive promotional impact on carbon productivity. Digitalization can realize low-carbon development and improve carbon productivity in manufacturing [34]. Similarly, manufacturing uses its collaboration in the process of servitization development to promote manufacturing production efficiency and enhance carbon productivity [35]. Therefore, further research is carried out to verify the influence of DSS on carbon productivity and explore the nonlinear relationship between DSS and carbon productivity.

**H1.** *The DSS exerts a positive influence on carbon productivity, and the two have nonlinear characteristics.*

*2.2. Conduction Path Analysis*

Industrial structure and technological innovation are important paths for manufacturing digitalization and servitization to reduce carbon emissions [15,29] and improve carbon productivity. Digitalization development can stimulate enterprises to change the traditional production model, develop a low-carbon model, and optimize industrial structure [36]. Through the upgrading of the enterprise's industrial structure, improvements in the level of technological innovation and carbon productivity can be achieved. [37]. The optimization of industrial structure and improvements at the level of technological innovation can improve resource utilization and reduce carbon emissions. They have a direct or indirect impact on enhancing carbon productivity [14]. The advancement of servitization fosters technological innovation within enterprises and optimizes their industrial structure, contributing positively to enhancing carbon productivity in manufacturing [38]. Optimizing industrial structure through the servitization approach has become a development trend in manufacturing. The substitution effect of service factors is utilized to reduce the input of energy factors and optimize the industrial structure of manufacturing. This provides an important impetus to decrease carbon emissions and bolster carbon productivity [39]. Manufacturing servitization improves the level of technological innovation in manufacturing through the input of knowledge factors. It leverages the positive externality of technological innovation to influence production activities positively, enhancing energy efficiency and carbon productivity [40].

**H2.** *Industrial structure and technological innovation serve as effective strategies for DSS to enhance carbon productivity.*

There are many research studies on economic growth and carbon emissions, but their conclusions are not consistent. Mujtaba and others [41] believe that there exists an inverse relationship between economic growth and carbon emissions, while Adjei and others [42] found a positive correlation between the two through empirical tests. Chen [43] empirically established an inverted "U" feature between economic growth and carbon emissions. However, Jiang [44] proposed that economic growth and carbon emissions do not exhibit an inverted "U" feature; instead, long-term changes with other shapes are exhibited. In the short term, economic growth raises carbon emissions at the cost of energy consumption. In the long term, economic growth contributes positively to reducing carbon emissions by changing the economic system and optimizing industrial structure [45]. Yu [17] verified that economic growth has a threshold effect on carbon emissions. He found that when economic growth exceeded the threshold, carbon emissions from enterprises relying on the development of highly polluting industries would increase. Li and Wang [46] discovered a positive influence between GDP and carbon productivity, which indicates that economic growth promotes carbon productivity. Wu and Yao [47] estimated economic growth patterns through economic growth and carbon productivity. They concluded that economic growth and carbon productivity had a two-way promoting effect. To sum up, economic growth significantly contributes to the enhancement of carbon productivity. We further verify the impact of DSS on carbon productivity with the development of the manufacturing economy when economic growth is the threshold variable.

**H3.** *There is a threshold effect in the relationship between DSS and carbon productivity, which results in a positive "U" relationship between DSS and carbon productivity.*

## 3. DSS Analysis for Manufacturing

### 3.1. The Coupling Coordination Degree

The coupling coordination degree measures the degree to which many elements within a system interact with each other. Based on the practices of Ma and others [48], by improving existing models, the DSS models of manufacturing industries in different provinces are proposed as follows:

$$C_{it} = 2\sqrt{u_{it} \times v_{it}} / (u_{it} + v_{it}) \tag{1}$$

In Equation (1), $C_{it}$ indicates the degree of coupling between digitalization and servitization in manufacturing, and a larger $C_{it}$ indicates better coupling between the two systems. $u_{it}$ is the digitalization score of province i in year t, and $v_{it}$ is the servitization score of province i in year t.

$$T_{it} = \alpha \times u_{it} + \beta \times v_{it} \tag{2}$$

$$DSS_{it} = \sqrt{C_{it} \times T_{it}} \tag{3}$$

The coupling coordination degree is a further confirmation of the relationship between digitalization and servitization in manufacturing, and it more accurately reflects the DSS in manufacturing for each province. In Equation (2), $\alpha$ and $\beta$ are coefficients to be determined; $\alpha$ represents the weighting in the manufacturing digitalization score and $\beta$ represents the weighting in the manufacturing servitization score: $\alpha + \beta = 1$ and $\alpha = \beta = 0.5$. In Equation (3), $DSS_{it}$ represents the degree of the DSS development of province i in year t.

Combining the criteria for classifying the levels of coupling coordination by Yang and others [49], this paper classifies manufacturing DSS into five types. The classification of DSS is shown in Table 1.

**Table 1.** DSS evaluation criteria.

| Range of Values | Grade | Level Identification | Synergy State Description |
|---|---|---|---|
| $0 \leq DSS < 0.2$ | Incongruity | E | No synergy, in a state of irrelevance, with a bias towards disorderly development |
| $0.2 \leq DSS < 0.4$ | Severe disorder | D | Lower level of synergy, in a haphazard state, entering a slow growth phase |
| $0.4 \leq DSS < 0.6$ | Primary coordination | C | General level synergy, in a state of loose partnership, entering an accelerated growth phase |
| $0.6 \leq DSS < 0.7$ $0.7 \leq DSS < 0.8$ | Moderate coordination | B | Medium–high-level synergy, in a state of healthy cooperation, entering a phase of rapid growth. |
| $0.8 \leq DSS < 0.9$ $0.9 \leq DSS \leq 1$ | Good coordination High-quality coordination | A | High synergy, in a highly cooperative state, entering a period of growth and mutation that will result in a new orderly structure |

*3.2. Index Selection and Data Source*

3.2.1. Manufacturing Digitalization

In this paper, we select the indicators related to the regional digitalization level and apply the entropy value method to obtain the scores of each indicator. Finally, the index value of manufacturing digitization level in 30 provinces of China is obtained, as shown in Tables 2 and 3.

**Table 2.** Manufacturing digitalization indicator system.

| Type | Indicator Description | Indicator Description |
|---|---|---|
| Digital Input | Number of enterprises with R&D activities [50] Number of enterprises with R&D activities [50] Percentage of businesses with e-commerce trading activities [51] R&D funding [50] Computers per 100 people [52] Number of websites per 100 businesses [52] | Reflects talent investment Reflects infrastructure investment |
| Digital Output | Revenue from sales of new products in manufacturing [53] Number of valid invention patents [53] Operating income [53] Unit energy consumption [54] Investment completed in industrial pollution control [55] | Reflects the output of technical and economic benefits Reflects the output of ecological benefits |

**Table 3.** Manufacturing digitalization level composite score.

| Year<br>Province | 2013 | 2014 | 2015 | 2016 | 2017 | 2018 | 2019 | 2020 |
|---|---|---|---|---|---|---|---|---|
| Beijing | 0.1096 | 0.1703 | 0.3774 | 0.3890 | 0.3527 | 0.3159 | 0.1629 | 0.2207 |
| Tianjin | 0.1729 | 0.5785 | 0.4991 | 0.2929 | 0.2418 | 0.1605 | 0.6212 | 0.2814 |
| Hebei | 0.2583 | 0.1682 | 0.4176 | 0.1529 | 0.0763 | 0.1438 | 0.3527 | 0.2435 |
| Shanxi | 0.1636 | 0.2065 | 0.0761 | 0.0361 | 0.0945 | 0.1757 | 0.0276 | 0.0346 |
| Inner Mongol | 0.0114 | 0.0165 | 0.0196 | 0.0241 | 0.0241 | 0.0211 | 0.0227 | 0.0273 |
| Liaoning | 0.0508 | 0.0586 | 0.0543 | 0.0538 | 0.0563 | 0.0622 | 0.0657 | 0.0734 |
| Jilin | 0.0114 | 0.0170 | 0.0201 | 0.0244 | 0.0266 | 0.0212 | 0.0228 | 0.0256 |
| Heilongjiang | 0.0124 | 0.0165 | 0.0168 | 0.0194 | 0.0208 | 0.0188 | 0.0228 | 0.0286 |
| Shanghai | 0.0887 | 0.1053 | 0.1127 | 0.1181 | 0.1241 | 0.1282 | 0.1398 | 0.1495 |
| Jiangsu | 0.2799 | 0.3387 | 0.3767 | 0.4124 | 0.4209 | 0.4437 | 0.5023 | 0.5426 |
| Zhejiang | 0.2091 | 0.2363 | 0.2635 | 0.2872 | 0.2951 | 0.3197 | 0.3665 | 0.4128 |
| Anhui | 0.0595 | 0.0821 | 0.0977 | 0.1126 | 0.1256 | 0.1312 | 0.1489 | 0.1652 |

**Table 3.** *Cont.*

| Province \ Year | 2013 | 2014 | 2015 | 2016 | 2017 | 2018 | 2019 | 2020 |
|---|---|---|---|---|---|---|---|---|
| Fujian | 0.0580 | 0.0714 | 0.0819 | 0.0961 | 0.1019 | 0.1141 | 0.1316 | 0.1492 |
| Jiangxi | 0.0205 | 0.0283 | 0.0379 | 0.0467 | 0.0561 | 0.0688 | 0.0853 | 0.1002 |
| Shandong | 0.1752 | 0.2057 | 0.2291 | 0.2644 | 0.2845 | 0.2858 | 0.2365 | 0.2869 |
| Henan | 0.0634 | 0.0791 | 0.0916 | 0.1036 | 0.1146 | 0.1188 | 0.1211 | 0.1337 |
| Hubei | 0.0576 | 0.0737 | 0.0877 | 0.1036 | 0.1070 | 0.1180 | 0.1315 | 0.1457 |
| Hunan | 0.0528 | 0.0670 | 0.0796 | 0.0937 | 0.1037 | 0.1221 | 0.1336 | 0.1483 |
| Guangdong | 0.2572 | 0.3003 | 0.3468 | 0.4054 | 0.4810 | 0.5294 | 0.5999 | 0.6575 |
| Guangxi | 0.0137 | 0.0176 | 0.0165 | 0.0222 | 0.0233 | 0.0258 | 0.0292 | 0.0348 |
| Hainan | 0.0102 | 0.0132 | 0.0139 | 0.0152 | 0.0158 | 0.0139 | 0.0145 | 0.0146 |
| Chongqing | 0.0223 | 0.0337 | 0.0429 | 0.0531 | 0.0611 | 0.0653 | 0.0699 | 0.0807 |
| Sichuan | 0.0397 | 0.0552 | 0.0655 | 0.0819 | 0.0902 | 0.0919 | 0.1052 | 0.1212 |
| Guizhou | 0.0048 | 0.0095 | 0.0133 | 0.0221 | 0.0248 | 0.0259 | 0.0277 | 0.0315 |
| Yunnan | 0.0092 | 0.0150 | 0.0213 | 0.0273 | 0.0292 | 0.0310 | 0.0379 | 0.0399 |
| Shanxi | 0.0232 | 0.0321 | 0.0364 | 0.0450 | 0.0505 | 0.0528 | 0.0587 | 0.0672 |
| Gansu | 0.0038 | 0.0077 | 0.0107 | 0.0121 | 0.0107 | 0.0106 | 0.0134 | 0.0148 |
| Qinghai | 0.0002 | 0.0017 | 0.0034 | 0.0047 | 0.0063 | 0.0090 | 0.0112 | 0.0124 |
| Ningxia | 0.0016 | 0.0040 | 0.0070 | 0.0086 | 0.0101 | 0.0118 | 0.0107 | 0.0146 |
| Xinjiang | 0.0026 | 0.0060 | 0.0081 | 0.0091 | 0.0086 | 0.0103 | 0.0112 | 0.0132 |

### 3.2.2. Manufacturing Servitization

To assess the extent of servitization in manufacturing, we draw on the measures of Li [56] and Oh [57] for empirical exploration. The DEA–Malmquist index is utilized to assess the evolution of total factor productivity in servitization within the manufacturing sector across 30 provinces in China over the period from 2013 to 2020. The relevant indicators of regional service levels were selected, as shown in Table 4. The scores of each index were obtained using the DEA–Malmquist index and the entropy value method, as shown in Table 5.

**Table 4.** Manufacturing servitization indicator system.

| Type | Indicator Description | References |
|---|---|---|
| Servicing Input Indicators | Selling costs [58] Management costs [58] Finance costs [58] | Reflect capital investment in marketing, after-sales, and others |
|  | R&D funding [59] R&D staff [59] | Reflect labor input in R&D, design, and others |
| Servicing Output Indicators | Operating income [60] Number of valid invention patents [59] | Reflect the service output of manufacturing |

**Table 5.** Manufacturing servitization level composite score.

| Province \ Year | 2013 | 2014 | 2015 | 2016 | 2017 | 2018 | 2019 | 2020 |
|---|---|---|---|---|---|---|---|---|
| Beijing | 0.6052 | 0.5791 | 0.9811 | 0.7431 | 0.7228 | 0.8505 | 0.6226 | 0.6488 |
| Tianjin | 0.4688 | 0.4340 | 0.7547 | 0.6386 | 0.7417 | 0.2235 | 0.6763 | 0.6284 |
| Hebei | 0.4848 | 0.3803 | 0.5370 | 0.4848 | 0.5356 | 0.5225 | 0.3672 | 0.4891 |

**Table 5.** *Cont.*

| Year / Province | 2013 | 2014 | 2015 | 2016 | 2017 | 2018 | 2019 | 2020 |
|---|---|---|---|---|---|---|---|---|
| Shanxi | 0.5428 | 0.5660 | 0.7634 | 0.3991 | 0.3614 | 0.6502 | 0.5907 | 0.5515 |
| Inner Mongol | 0.3817 | 0.5414 | 0.4615 | 0.4267 | 0.9536 | 0.2250 | 0.6328 | 0.5269 |
| Liaoning | 0.5152 | 0.6763 | 0.6488 | 0.3687 | 0.2729 | 0.5689 | 0.6183 | 0.7663 |
| Jilin | 0.5065 | 0.5254 | 0.5414 | 0.4485 | 0.7678 | 0.9086 | 0.0595 | 0.5022 |
| Heilongjiang | 0.4412 | 0.5588 | 0.5559 | 0.5385 | 0.6894 | 0.3280 | 0.5820 | 0.7083 |
| Shanghai | 0.2772 | 0.5849 | 0.6110 | 0.9637 | 0.5167 | 0.8171 | 0.8766 | 0.4136 |
| Jiangsu | 0.4775 | 0.3904 | 0.6633 | 0.6168 | 0.6792 | 0.6372 | 0.3991 | 0.4107 |
| Zhejiang | 0.4993 | 0.3991 | 0.6386 | 0.4877 | 0.5646 | 0.5152 | 0.5893 | 0.6226 |
| Anhui | 0.6807 | 0.6357 | 0.7460 | 0.6865 | 0.5936 | 0.6009 | 0.3759 | 0.5864 |
| Fujian | 0.4673 | 0.4412 | 0.7547 | 0.6546 | 0.5893 | 0.5559 | 0.6313 | 0.6415 |
| Jiangxi | 0.5080 | 0.4209 | 0.6734 | 0.4514 | 0.5428 | 0.4020 | 0.2946 | 0.5588 |
| Shandong | 0.5530 | 0.5399 | 0.6865 | 0.5791 | 0.6226 | 0.6226 | 0.2714 | 0.3309 |
| Henan | 0.4673 | 0.4804 | 0.7242 | 0.5225 | 0.5791 | 0.8433 | 0.0000 | 0.6575 |
| Hubei | 0.4949 | 0.5718 | 0.7286 | 0.6357 | 0.6749 | 0.5225 | 0.4049 | 0.7576 |
| Hunan | 0.5791 | 0.5573 | 0.6705 | 0.5007 | 0.6604 | 0.6531 | 0.3570 | 0.5515 |
| Guangdong | 0.5617 | 0.5965 | 0.9086 | 0.9086 | 0.6343 | 0.7504 | 0.4906 | 0.5646 |
| Guangxi | 0.6139 | 0.5922 | 0.8897 | 0.6415 | 0.6023 | 0.7997 | 0.2496 | 0.3861 |
| Hainan | 0.6212 | 0.7460 | 0.6734 | 1.0000 | 0.5007 | 0.2380 | 0.7765 | 0.5849 |
| Chongqing | 0.4978 | 0.3425 | 0.6691 | 0.6067 | 0.9042 | 0.4557 | 0.3556 | 0.5922 |
| Sichuan | 0.5022 | 0.6880 | 0.5486 | 0.5457 | 0.6444 | 0.5167 | 0.3774 | 0.5573 |
| Guizhou | 0.7358 | 0.7068 | 0.6821 | 0.5443 | 0.7141 | 0.4122 | 0.2961 | 0.5080 |
| Yunnan | 0.6212 | 0.5776 | 0.6096 | 0.4586 | 0.3962 | 0.4964 | 0.6604 | 0.4819 |
| Shanxi | 0.5254 | 0.4775 | 0.6328 | 0.5994 | 0.6226 | 0.8534 | 0.4659 | 0.7184 |
| Gansu | 0.6589 | 0.6328 | 0.6604 | 0.5109 | 0.4557 | 0.6952 | 0.5327 | 0.4369 |
| Qinghai | 0.7605 | 0.5327 | 0.9913 | 0.2482 | 0.5631 | 0.7199 | 0.3135 | 0.7228 |
| Ningxia | 0.5864 | 0.6589 | 0.5747 | 0.4165 | 0.4906 | 0.6255 | 0.6851 | 0.7286 |
| Xinjiang | 0.5530 | 0.5370 | 0.6502 | 0.4122 | 0.6096 | 0.8839 | 0.7678 | 0.6531 |

#### 3.2.3. Data Source

The data in this paper cover the period 2013–2020. The main sources of data are the China Statistical Yearbook (2014–2021), China Industry Statistical Yearbook (2014–2021), China Energy Statistical Yearbook (2014–2021), China Statistical Yearbook on Science and Technology (2014–2021), National Bureau of Statistics, and statistical yearbooks for each region. Xizang has been excluded from the measurement due to more serious data deficiencies, while Hong Kong, Macau, and Taiwan are not included.

### 3.3. The Manufacturing DSS

According to the development level of the digitalization and servitization of manufacturing measured in the previous section, this study used the coupling coordination degree model to calculate DSS for the 30 provinces of China and their coordination levels, as shown in Table 6. The specific trends are shown in Figure 1.

**Table 6.** Degree of DSS and level of coordination in manufacturing.

| Province \ Year | 2013 | 2014 | 2015 | 2016 | 2017 | 2018 | 2019 | 2020 |
|---|---|---|---|---|---|---|---|---|
| Beijing | 0.5075 (C) | 0.5604 (C) | 0.7801 (B) | 0.7332 (B) | 0.7106 (B) | 0.7199 (B) | 0.5644 (C) | 0.6151 (B) |
| Tianjin | 0.5336 (C) | 0.7078 (B) | 0.7834 (B) | 0.6577 (B) | 0.6507 (B) | 0.4352 (C) | 0.8051 (A) | 0.6485 (B) |
| Hebei | 0.5949 (C) | 0.5029 (C) | 0.6881 (B) | 0.5218 (C) | 0.4497 (C) | 0.5235 (C) | 0.5999 (C) | 0.5874 (C) |
| Shanxi | 0.5459 (C) | 0.5847 (C) | 0.4909 (C) | 0.3464 (D) | 0.4298 (C) | 0.5814 (C) | 0.3574 (D) | 0.3716 (D) |
| Inner Mongol | 0.2570 (D) | 0.3072 (D) | 0.3083 (D) | 0.3184 (D) | 0.3894 (D) | 0.2626 (D) | 0.3463 (D) | 0.3465 (D) |
| Liaoning | 0.4021 (C) | 0.4461 (C) | 0.4332 (C) | 0.3753 (D) | 0.3520 (D) | 0.4337 (C) | 0.4490 (C) | 0.4870 (C) |
| Jilin | 0.2754 (D) | 0.3074 (D) | 0.3230 (D) | 0.3233 (D) | 0.3780 (D) | 0.3726 (D) | 0.1919 (F) | 0.3366 (D) |
| Heilongjiang | 0.2718 (D) | 0.3100 (D) | 0.3109 (D) | 0.3196 (D) | 0.3460 (D) | 0.2804 (D) | 0.3393 (D) | 0.3773 (D) |
| Shanghai | 0.3960 (D) | 0.4982 (C) | 0.5123 (C) | 0.5809 (C) | 0.5032 (C) | 0.5689 (C) | 0.5917 (C) | 0.4987 (C) |
| Jiangsu | 0.6046 (B) | 0.6030 (B) | 0.7070 (B) | 0.7102 (B) | 0.7312 (B) | 0.7292 (B) | 0.6691 (B) | 0.6871 (B) |
| Zhejiang | 0.5684 (C) | 0.5542 (C) | 0.6405 (B) | 0.6118 (B) | 0.6389 (B) | 0.6371 (B) | 0.6817 (B) | 0.7120 (B) |
| Anhui | 0.4486 (C) | 0.4780 (C) | 0.5195 (C) | 0.5273 (C) | 0.5226 (C) | 0.5299 (C) | 0.4864 (C) | 0.5579 (C) |
| Fujian | 0.4057 (C) | 0.4213 (C) | 0.4987 (C) | 0.5008 (C) | 0.4951 (C) | 0.5019 (C) | 0.5369 (C) | 0.5562 (C) |
| Jiangxi | 0.3193 (D) | 0.3304 (D) | 0.3996 (D) | 0.3811 (D) | 0.4178 (C) | 0.4078 (C) | 0.3981 (D) | 0.4864 (C) |
| Shandong | 0.5579 (C) | 0.5773 (C) | 0.6298 (B) | 0.6256 (B) | 0.6488 (B) | 0.6495 (B) | 0.5034 (C) | 0.5551 (C) |
| Henan | 0.4150 (C) | 0.4415 (C) | 0.5075 (C) | 0.4824 (C) | 0.5076 (C) | 0.5626 (C) | 0.0079 (E) | 0.5445 (C) |
| Hubei | 0.4108 (C) | 0.4530 (C) | 0.5028 (C) | 0.5066 (C) | 0.5184 (C) | 0.4983 (C) | 0.4804 (C) | 0.5764 (C) |
| Hunan | 0.4182 (C) | 0.4396 (C) | 0.4807 (C) | 0.4654 (C) | 0.5115 (C) | 0.5314 (C) | 0.4673 (C) | 0.5348 (C) |
| Guangdong | 0.6165 (B) | 0.6506 (B) | 0.7492 (B) | 0.7790 (B) | 0.7432 (B) | 0.7939 (B) | 0.7365 (B) | 0.7806 (B) |
| Guangxi | 0.3026 (D) | 0.3197 (D) | 0.3478 (D) | 0.3456 (D) | 0.3440 (D) | 0.3788 (D) | 0.2922 (D) | 0.3404 (D) |
| Hainan | 0.2823 (D) | 0.3149 (D) | 0.3113 (D) | 0.3509 (D) | 0.2984 (D) | 0.2396 (D) | 0.3258 (D) | 0.3038 (D) |
| Chongqing | 0.3246 (D) | 0.3279 (D) | 0.4115 (C) | 0.4237 (C) | 0.4849 (C) | 0.4153 (C) | 0.3971 (D) | 0.4675 (C) |
| Sichuan | 0.3757 (D) | 0.4415 (C) | 0.4354 (C) | 0.4597 (C) | 0.4911 (C) | 0.4669 (C) | 0.4464 (C) | 0.5098 (C) |
| Guizhou | 0.2437 (D) | 0.2863 (D) | 0.3085 (D) | 0.3311 (D) | 0.3648 (D) | 0.3215 (D) | 0.3010 (D) | 0.3558 (D) |
| Yunnan | 0.2749 (D) | 0.3050 (D) | 0.3374 (D) | 0.3345 (D) | 0.3280 (D) | 0.3522 (D) | 0.3977 (D) | 0.3725 (D) |
| Shanxi | 0.3323 (D) | 0.3519 (D) | 0.3895 (D) | 0.4054 (C) | 0.4210 (C) | 0.4608 (C) | 0.4067 (C) | 0.4687 (C) |
| Gansu | 0.2240 (D) | 0.2642 (D) | 0.2898 (D) | 0.2805 (D) | 0.2646 (D) | 0.2931 (D) | 0.2908 (D) | 0.2838 (D) |
| Qinghai | 0.1055 (E) | 0.1745 (E) | 0.2409 (D) | 0.1852 (F) | 0.2441 (D) | 0.2837 (D) | 0.2432 (D) | 0.3079 (D) |
| Ningxia | 0.1741 (E) | 0.225 (D) | 0.2520 (D) | 0.2448 (D) | 0.2656 (D) | 0.2930 (D) | 0.2923 (D) | 0.3213 (D) |
| Xinjiang | 0.1956 (E) | 0.2385 (D) | 0.2691 (D) | 0.2472 (D) | 0.2690 (D) | 0.3087 (D) | 0.3046 (D) | 0.3047 (D) |

By calculating manufacturing DSS in 30 provinces of China, the degree of the manufacturing DSS was measured. As depicted in Table 4, the manufacturing DSS across 30 provinces in China tends to rise from 2013 to 2020, but the overall level is low and in a state of imbalance. The manufacturing DSS of most provinces is maintained in a state of loose cooperation and low coordination, but the overall DSS comprises a stage of growth.

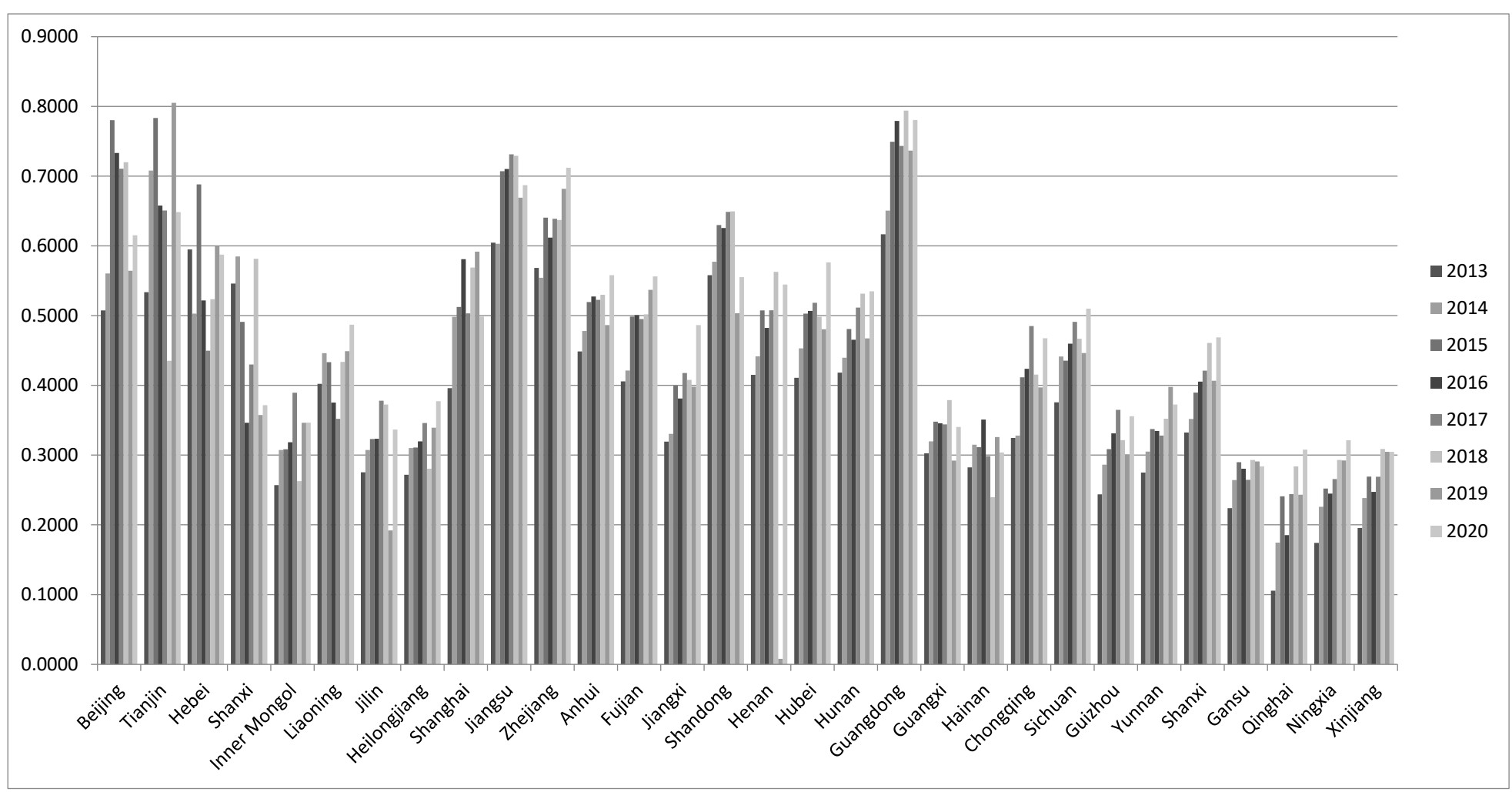

**Figure 1.** Manufacturing DSS degree of 30 provinces in China, 2013–2020.

In terms of distribution (Figure 2), Beijing, Tianjin, Jiangsu, Guangdong, and other economically developed regions have a good economic foundation and technical support, and DSS is relatively high. Their digitalization and servitization coupling coordination degree is good, and they are in the general coordination stage. The DSS in Hebei, Shandong, Sichuan, Shaanxi, and other regions is relatively low, and it is at a low level of coordination. DSS relationships tend to develop in an orderly manner, but there is still a certain distance to the high-level goal. The DSS in Gansu, Yunnan, Xinjiang, and other regions is at a lower level as it is limited by economic and technological constraints, and the DSS is lower than that in other regions. Therefore, while emphasizing rapid economic development, manufacturing needs to pay special attention to injecting digital capabilities and service capabilities into the industry.

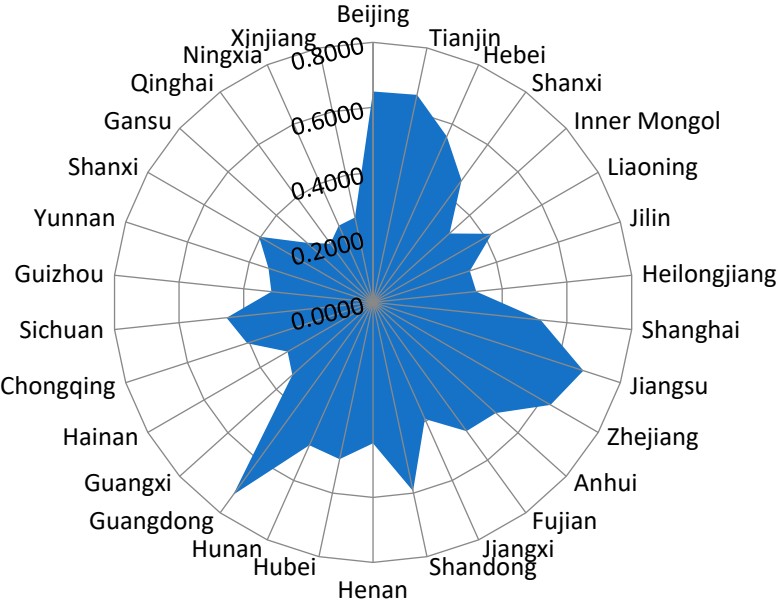

**Figure 2.** Average level of DSS in manufacturing, 2013–2020.

In general, the collaborative development of digitalization and servitization in manufacturing is in the early stages, and it comprises good resources and industrial and sustainable development. However, the development of DSS in manufacturing is a long-term process, and the average DSS level within China's manufacturing is far from a high level of synergy. Hence, there is a need to expedite the coordinated advancement of digitalization and servitization to a greater extent.

## 4. Empirical Study on The Impact of DSS on Carbon Productivity

### 4.1. Model Building

To explore the influence of DSS on carbon productivity in manufacturing, we have constructed the following regression model, which is built upon the extended STIRPAT model [61]:

$$CP_{it} = \alpha_0 + \alpha_1 DSS_{it} + \beta X_{it} + \gamma_{year} + \gamma_{pro} + \theta_{it} \tag{4}$$

In Equation (4), $CP_{it}$ is the carbon productivity of province i in year t. $DSS_{it}$ is the DSS score of province i in year t. $X_{it}$ represents the control variables, including administrative control, infrastructure development, profit performance, enterprise scale, and industry debt ratios. $\gamma_{year}$ and $\gamma_{pro}$ are the fixed effects of the province and time, and $\theta_{it}$ is an error term.

The above theories show that both technological innovation and industrial structure exert a notable influence on carbon productivity in manufacturing. This paper further draws on relevant studies on mediating effect tests [62] to set up a mediating effect test model:

$$CP_{it} = \alpha_0 + \alpha_1 DSS_{it} + \beta X_{it} + \gamma_{year} + \gamma_{pro} + \theta_{it} \tag{5}$$

$$M_{it} = \alpha_0 + \alpha_1 DSS_{it} + \beta X_{it} + \gamma_{year} + \gamma_{pro} + \theta_{it} \tag{6}$$

$$CP_{it} = \alpha_0 + \alpha_1 DSS_{it} + \alpha_2 M_{it} + \beta X_{it} + \gamma_{year} + \gamma_{pro} + \theta_{it} \tag{7}$$

In Equations (6) and (7), $M_{it}$ is the mediating variable, including technological innovation and industrial structure.

To examine the threshold effect between DSS and carbon productivity, we utilize the threshold regression model proposed by Hansen [63]. The threshold effect test model is described as follows:

$$CP_{it} = \alpha_0 + \alpha_1 DSS_{it} \times I(q_{it} \leq \varepsilon_1) + \alpha_2 DSS_{it} \times I(q_{it} > \varepsilon_1) + \beta X_{it} + \gamma_{year} + \gamma_{pro} + \theta_{it} \tag{8}$$

In Equation (8), $q_{it}$ is the threshold variable, and $\varepsilon_1$ is the threshold value.

### 4.2. Variable Selection

The explained variable comprises carbon productivity (CP). According to the measure of KAYA [64], carbon productivity is used to measure carbon productivity in manufacturing, which is expressed as the ratio of sales revenue to carbon emissions in manufacturing.

The core explanatory variables comprise the synergy of digitalization and servitization (DSS). The DSS is used as the core explanatory variable.

The mediating variables comprise technological innovation (sit) [65], measured as a ratio of the number of valid invention patents to R&D expenditure in manufacturing in each provincial administrative region. Industrial structure (str) [66] is expressed by adopting the ratio of the added value of the tertiary industry to the added value of the secondary industry.

The threshold variable, economic growth (income) [33], is expressed as the logarithm of manufacturing operating income.

Control variables were included to reduce model estimation bias, and five control variables were included in the empirical analysis.

Administrative control (gov) [67] is indicated by the share of government fiscal expenditure to GDP.

The enterprise scale (scal) [58] is expressed as the ratio of the main business income to the number of business units in manufacturing.

Infrastructure development (tra) [68] is measured using the logarithm of the total number of bus passengers traveling in the city at the year's end.

Profit performance (pro) [69] is selected to express the share of profits relative to the main business revenue in each province.

Industry debt ratios (idebt) [70] are selected as the ratio of total liabilities to total assets for manufacturing.

### 4.3. Result Analysis

#### 4.3.1. Descriptive Statistics

In Table 7, carbon productivity in manufacturing varies widely across provinces. The mean of CP is 0.976, the minimum of CP is 0.163, and the maximum of CP is 3.678. This indicates that overall carbon productivity in manufacturing needs to be improved. Manufacturing DSS has a standard deviation of 0.151 and minimum and maximum values of 0.008 and 0.803, respectively. This shows that the DSS level in manufacturing exhibits substantial variations in different regions. The mean of DSS is 0.441, which indicates that the vast majority of provinces have their DSS at a low level and that the DSS in manufacturing in most regions has more room for development.

**Table 7.** Descriptive statistics.

| Variable | N | Mean | Std.Dev. | Min | Max |
|---|---|---|---|---|---|
| CP | 240 | 0.976 | 0.671 | 0.163 | 3.679 |
| DSS | 240 | 0.441 | 0.151 | 0.008 | 0.805 |

4.3.2. Correlation Analysis

The Pearson correlation coefficient was used to examine the relationship between variables, and the outcomes are presented in Table 8. The results show that DSS and CP are positively correlated. This is a preliminary indication that DSS positively affects CP in manufacturing, and it lays the foundation for investigating the intrinsic link between the two.

**Table 8.** Correlation analysis.

| | CP | DSS | scal | idebt | pro | gov | tra |
|---|---|---|---|---|---|---|---|
| CP | 1 | | | | | | |
| DSS | 0.506 *** | 1 | | | | | |
| scal | −0.123 * | −0.063 | 1 | | | | |
| idebt | −0.644 *** | −0.375 *** | 0.007 | 1 | | | |
| pro | 0.226 *** | 0.126 * | −0.037 | −0.437 *** | 1 | | |
| gov | −0.519 *** | −0.600 *** | 0.306 *** | 0.491 *** | −0.246 *** | 1 | |
| tra | 0.361 *** | 0.530 *** | −0.365 *** | −0.304 *** | 0.142 ** | −0.701 *** | 1 |

Note: *, **, and *** denote significance levels of 10%, 5%, and 1%, respectively.

4.3.3. Panel Data Model Selection

The model used needs to be screened before conducting the empirical test. As shown in Table 9, the Hausman test is 678.42, and the *p*-value is 0.0000. As a result of this correlation, the fixed-effect model was ultimately chosen for adoption, and the *p*-values of the F-test statistics were all 0.00, so the two-way fixed-effect model was chosen.

**Table 9.** Panel data model selection: test results.

| Test Methods | Time Effect | | Individual Effect | | Double Effect | | chi2() | Prob > chi2() |
|---|---|---|---|---|---|---|---|---|
| | F-Statistics | Prob > F | F-Statistics | Prob > F | F-Statistics | Prob > F | | |
| Hausman Test | | | | | | | 678.42 | 0.00 |
| F Test | 15.69 | 0.00 | 15.69 | 0.00 | 23.81 | 0.00 | | |

4.3.4. Basic Regression Analysis

This paper conducts regression analyses using panel data for 30 Chinese provinces spanning the period 2013–2020. The sample is further analyzed by dividing the sample into the East (China's East region includes 12 provinces, autonomous regions, and municipalities directly under the Central Government, including Beijing, Tianjin, Hebei, Liaoning, Shanghai, Jiangsu, Zhejiang, Fujian, Shandong, Guangdong, Guangxi, and Hainan) and the Midwest (China's Midwest region includes 19 provinces and autonomous regions, including Shanxi, Inner Mongolia, Jilin, Heilongjiang, Anhui, Jiangxi, Henan, Hubei, Hunan, Sichuan, Chongqing, Guizhou, Yunnan, Tibet, Shaanxi, Gansu, Ningxia, Qinghai, and Xinjiang. This study does not include the Tibet Autonomous Region) according to the classification of the Ministry of Finance's "Opinions on Clarifying the Division of Eastern, Midwestern and Western Regions".

Column (1) in Table 10 shows that DSS has a significant positive relationship with CP in manufacturing. Column (2) is a double fixed-effect regression controlling for time and province. The results in column (2) show a positive correlation between DSS and carbon productivity in China, the East, and the Midwest sample levels. This suggests that DSS has a significant contribution to CP in manufacturing. The effect of enterprise scales on

CP is not significant, and the industrial debt ratios has a significant negative influence on CP in manufacturing in the East. The result shows that during the development of DSS in manufacturing, the input of DSS elements reduces energy resource inputs. This reduces carbon emissions, raises manufacturing revenues, and effectively increases manufacturing carbon productivity. By region, each 1% increase in manufacturing DSS in the East is associated with a 0.75% increase in CP, while the influence of DSS on CP is insignificant in the Midwest. The East has abundant human capital elements and technological elements, which guarantee the input of economic, technological, and service elements in the East. These resources strongly support the development of DSS in manufacturing in the East and promote the CP of manufacturing. Manufacturing in the Midwest is still mostly resource-dependent, with insufficient inputs of service factors, and the level of technology and human capital needs to be upgraded. Thus, a high degree of manufacturing DSS is not present, and the Midwest has not yet paid dividends in terms of increased CP in manufacturing [68].

**Table 10.** Fundamental regression results.

| Variables | Full Sample | | The East | | The Midwest | | U-Shaped Inspection | U-Shaped Inspection |
|---|---|---|---|---|---|---|---|---|
| | (1) CP | (2) CP | (1) CP | (2) CP | (1) CP | (2) CP | (3) CP | (4) CP |
| DSS | 2.224 *** | 0.265 ** | 1.953 *** | 0.750 * | 0.992 *** | 0.116 | −0.859 ** | −1.238 *** |
| | (11.64) | (2.512) | (5.471) | (1.861) | (3.610) | (0.933) | (−2.293) | (−3.602) |
| $DSS^2$ | | | | | | | 1.606 *** | 1.990 *** |
| | | | | | | | (3.321) | (4.575) |
| scal | −0.126 ** | 0.0136 | −0.176 | 0.0416 | −0.118 | −0.0111 | | 0.0136 |
| | (−2.093) | (0.300) | (−1.639) | (0.834) | (−1.538) | (−0.183) | | (0.316) |
| idebt | −3.411 *** | −0.536 * | −4.321 *** | −0.533 | −3.182 *** | −0.259 | | −0.634 ** |
| | (−8.476) | (−1.736) | (−5.071) | (−0.416) | (−6.994) | (−0.647) | | (−2.146) |
| pro | −0.342 | −0.243 | 3.394 | 6.186 ** | −1.609 ** | −0.515 | | −0.575 * |
| | (−0.466) | (−0.778) | (1.554) | (2.342) | (−2.264) | (−1.492) | | (−1.873) |
| gov | 0.699 *** | −1.401 *** | −0.317 | −6.347 *** | 0.563 * | −1.237 *** | | −1.491 *** |
| | (2.642) | (−6.456) | (−0.462) | (−4.353) | (1.956) | (−5.110) | | (−7.179) |
| tra | −0.434 ** | −0.973 *** | −0.340 | −1.158 *** | −0.0585 | 0.674 | | −0.664 ** |
| | (−2.554) | (−3.116) | (−1.104) | (−3.644) | (−0.274) | (1.286) | | (−2.178) |
| Constant | 2.289 *** | 3.005 *** | 2.674 *** | 9.561 *** | 1.911 *** | −0.371 | 0.715 *** | 2.745 *** |
| | (5.517) | (4.839) | (3.084) | (5.376) | (3.879) | (−0.345) | (9.834) | (4.617) |
| Observations | 240 | 240 | 88 | 88 | 152 | 152 | 240 | 240 |
| R-squared | 0.643 | 0.979 | 0.646 | 0.959 | 0.445 | 0.966 | 0.973 | 0.981 |
| TE | | YES | | YES | | YES | YES | YES |
| FE | | YES | | YES | | YES | YES | YES |

Note: *, **, and *** denote significance levels of 10%, 5%, and 1%, respectively; t-values in parentheses.

Columns (3) and (4) show the nonlinear effect of DSS on CP for the two cases in the existence and nonexistence of control variables. The findings of the study indicate that the coefficient of the primary term of DSS is negative at the 5% significance level, while the coefficient of its quadratic term is positive at the 1% significance level, regardless of whether the control variable is included or not. This reveals a positive "U" association between DSS and CP. Upon further inspection, the extreme point of the positive "U" relationship in column (4) is 0.311, and the values of DSS are in the range of [0.161, 0.579]. It can be observed that the turning point lies within the range of the independent variable, and the initial hypothesis is rejected at the 1% significance level, confirming the validity of the test. A positive "U" relationship between DSS and CP is established, and there is a nonlinear relationship between the two. As a result, H1 is valid.

### 4.3.5. Robustness and Endogeneity

To alleviate the endogeneity problem of the model, the first-order lag of the DSS is used. Column (1) in Table 11 demonstrates that the coefficient of the first-order lag of the DSS is 0.184, with a significance level of 10%, thereby confirming the robustness of the baseline regression results.

**Table 11.** Robustness tests.

| Variables | CP (1) | CP (2) |
|---|---|---|
| L.CP | | 1.095 *** |
| | | (50.11) |
| L.DSS | 0.184 * | |
| | (1.863) | |
| DSS | | 0.270 *** |
| | | (3.242) |
| Control Variable | YES | YES |
| Constant | 2.965 *** | 0.120 |
| | (4.107) | (0.798) |
| Observations | 210 | 210 |
| R-squared | 0.983 | |
| TE | YES | YES |
| FE | YES | YES |
| AR(1)($p$-value) | | 0.029 |
| AR(2) ($p$-value) | | 0.117 |
| Hansen ($p$-value) | | 0.078 |

Note: * and *** denote significance levels of 10% and 1%, respectively; *t*-values in parentheses.

Considering the model's autocorrelation, a dynamic panel with the first-order lag of CP was constructed using the systematic GMM model for regression tests. In model (2), the regression results from the GMM model indicate that DSS continues to have a significant promoting effect on CP, suggesting that the baseline results remain robust.

### 4.3.6. Intermediary Effects Test

This paper further studies the impact of DSS on CP in manufacturing through two factors: technological innovation and industrial structure. Columns (2) and (3) of Table 12 demonstrate a notable positive impact of DSS on industrial structure, and DSS has a non-significant relationship with technological innovation. Column (4) and column (5) show that on the basis of basic regression, with the addition of the mediator variable, DSS and the mediator variable still have a contributing effect on CP. The boosting effect remains significantly positive at the 1% level. This suggests a mediating effect. Comparing column (1), the coefficients in columns (4) and (5) drop to 0.268 and 0.281 when technological innovation and industrial structure are added. This indicates that technological innovation and industrial structure are mediators in the influence of DSS on CP. The improvements in industrial structure and technological innovation can improve the effect of DSS on CP. As a result, H2 is valid.

**Table 12.** Intermediary effect test.

| Variables | CP (1) | sit (2) | str (3) | CP (4) | CP (5) |
|---|---|---|---|---|---|
| DSS | 0.334 *** | −0.478 | 0.184 ** | 0.268 *** | 0.281 *** |
| | (3.067) | (−1.530) | (2.062) | (2.658) | (2.620) |
| sit | | | | −0.138 *** | |
| | | | | (−6.098) | |
| str | | | | | 0.290 *** |
| | | | | | (3.461) |
| Constant | 0.510 *** | 4.231 *** | 0.564 *** | 1.093 *** | 0.347 *** |
| | (12.93) | (37.40) | (17.46) | (10.69) | (5.700) |
| Observations | 240 | 240 | 240 | 240 | 240 |
| R-squared | 0.972 | 0.916 | 0.931 | 0.976 | 0.973 |
| TE | YES | YES | YES | YES | YES |
| FE | YES | YES | YES | YES | YES |

Note: **, and *** denote significance levels of 5%, and 1%, respectively; t-values in parentheses.

Table 13 shows the findings of the Soble test for mediating effects. According to the Z-statistic, Z is significant at the 0.05% level, so the Soble test passes the hypothesis of mediating effects. The indirect effects of DSS on carbon productivity through technological innovation and industrial structure are 0.166 and 0.120, respectively. The direct effects were 1.959 and 2.005, respectively. The intermediary effects accounted for 7.79% and 5.63%, respectively. This shows that technological innovation and industrial structure are important ways for DSS to improve CP. As a result, H2 is again valid.

**Table 13.** Soble test results.

| Intermediate Variables | Indirect Effects | Direct Effects | Total Effect | Percentage of Intermediary Effect | Z Statistic |
|---|---|---|---|---|---|
| sit | 0.166 *** | 1.959 *** | 2.125 *** | 7.79% | 3.071 *** |
| str | 0.120 ** | 2.005 *** | 2.125 *** | 5.63% | 2.392 ** |

Note: **, and *** denote significance levels of 5%, and 1%, respectively.

4.3.7. Threshold Effect Analysis

Through the above tests, it is evident that there exists a positive "U" relationship between DSS and CP. This paper takes DSS as the independent variable and DSS and economic growth as the threshold variables to conduct the threshold test.

Table 14 displays the test's outcomes. We find that the test results for the single threshold of the DSS are significant at the 1% level. Nonetheless, the double and triple thresholds do not show significance. Hence, there exists a single threshold for DSS with an estimated threshold of 0.2383. Single threshold regression is used for the next analysis. The tests for economic growth indicate significance for both the single and double thresholds, while the triple threshold test is non-significant. Therefore, there are single and double thresholds for economic growth, and these thresholds are 6.5937 and 10.8619, respectively. Double threshold regression was used for further analysis.

The threshold effect was tested by drawing the likelihood ratio function graph of DSS and economic growth. The likelihood ratio function was used to express the relationship between LR values and threshold values. As shown in Figures 3 and 4, the 95% confidence interval for the threshold estimate is the interval formed by the critical value of 7.35 (corresponding to the dotted line in Figures 3 and 4) for all LR values less than the 5% significance level. It indicates that the threshold value of threshold regression is equivalent to the actual threshold value, which is in agreement with the findings from the earlier significance test.

**Table 14.** Test results of the threshold effect.

| Independent Variable | Threshold Variable | Hypothesis Testing | RSS | MSE | F-Statistics | *p*-Value | Threshold Value | 95% Confidence Interval |
|---|---|---|---|---|---|---|---|---|
| DSS | DSS | single threshold | 0.5788 | 0.0025 | 36.86 *** | 0.0000 | 0.2383 | |
| | | double threshold | 0.5437 | 0.0023 | 14.96 | 0.1133 | 0.3471 | [0.2350, 0.2433] |
| | | triple threshold | 0.5230 | 0.0023 | 9.20 | 0.6033 | 0.5011 | |
| DSS | income | single threshold | 0.6019 | 0.0026 | 26.53 ** | 0.0267 | 6.5937 | [5.9234, 7.1117] |
| | | double threshold | 0.5567 | 0.0024 | 18.86 ** | 0.0500 | 10.8619 | [10.7945, 11.0311] |
| | | triple threshold | 0.5417 | 0.0023 | 6.42 | 0.7633 | 11.2032 | |

Note: **, and *** denote significance levels of 5%, and 1%, respectively.

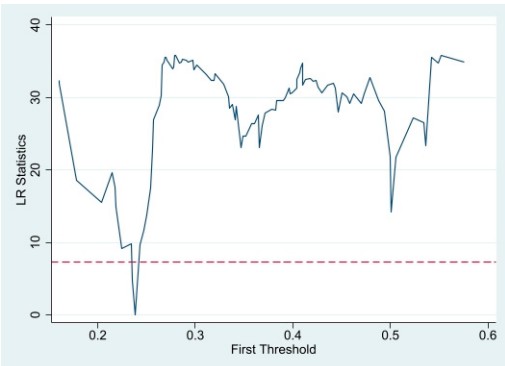

**Figure 3.** Single threshold estimates and 95% confidence intervals for DSS.

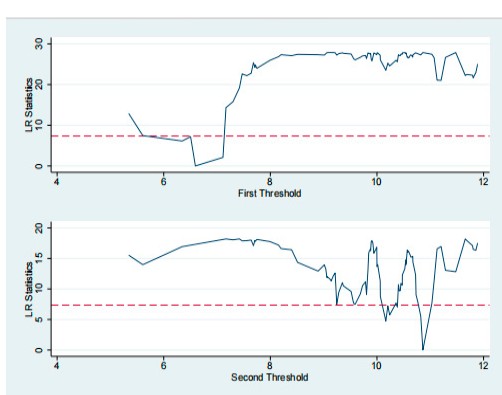

**Figure 4.** Dual threshold estimates and 95% confidence intervals for economic growth.

We conducted regression analyses using the threshold model, and the outcomes are presented in Table 15.

Column (1) shows that when DSS is used as the threshold variable, a significant positive correlation exists between DSS and CP at the 1% significance level when DSS < 0.2383, with a coefficient of 0.864. When DSS ≥ 0.2383, DSS and CP are still significantly positively correlated at the 1% level, but the coefficient is 0.418, and its contribution decreases as DSS increases. Currently, most of the DSS in manufacturing is in the low-synergy stage, and the effect of DSS on CP does not reach a higher level of facilitation for the time being. The effect is differentiated by the development stage of DSS. Therefore, there is a nonlinear positive correlation between DSS and CP, and H1 is again valid.

**Table 15.** Parameter estimation results of the threshold regression model.

| Dependent Variable | CP (1) | CP (2) |
| --- | --- | --- |
| Independent Variable | DSS | DSS |
| Threshold Variable | DSS | income |
| DSS (DSS < $\theta_1$) | 0.864 *** | |
| | (5.741) | |
| DSS (DSS $\geq$ $\theta_1$) | 0.418 *** | |
| | (3.160) | |
| DSS (income < $\gamma_1$) | | −0.073 |
| | | (−0.482) |
| DSS ($\gamma_1 \leq$ income < $\gamma_2$) | | 0.250 ** |
| | | (2.251) |
| DSS (income $\geq$ $\gamma_2$) | | 0.059 |
| | | (0.452) |
| Control Variable | YES | YES |
| Constant | −1.907 *** | −1.768 *** |
| | (−4.476) | (−4.209) |
| Observations | 240 | 240 |
| R-squared | 0.677 | 0.687 |

Note: **, and *** denote significance levels of 5%, and 1%, respectively; *t*-values in parentheses.

Column (2) demonstrates that when economic growth is taken as the threshold variable, there is a negative correlation between DSS and CP when economic growth is <6.5937, with a coefficient of −0.073. When economic growth is at [6.5937, 10.8619], DSS was significantly and positively correlated with CP at 5%. When economic growth $\geq$ 10.8619, DSS has a positive correlation with CP, but its promoting effect is not significant. Therefore, with the economic growth of manufacturing, DSS has a positive "U" relationship with CP. Economic growth has a threshold effect on the impact of DSS on CP, and H3 is valid.

## 5. Discussion

This paper empirically analyzes the influence of manufacturing DSS on carbon productivity in 30 provinces of China from 2013 to 2020 and divides manufacturing DSS levels. The conclusion is as follows.

First, the level of DSS in China's manufacturing is low, and the average level is still far from a high level of cooperation. China's manufacturing DSS faces the challenge of uneven development, with obvious regional differences [71]. In developed eastern regions like Beijing, Guangdong, and other regions, the level of DSS implementation surpasses that in less developed areas, yet it still has a certain gap from a high level of coordination.

Second, DSS makes a notable positive contribution to carbon productivity, and the two show a strong and then weak non-linear relationship. The reason is that the development of DSS relies on electricity, computing power, high and new technologies, and others [22]. These supporting factors require a large amount of energy consumption to a certain extent, and the technology development cycle is long, which will hinder the contribution of DSS to carbon productivity in the short term. Currently, DSS in most provinces is at a low level; DSS in developed provinces such as Beijing and Guangdong is only at a general level, and it has not reached a high level of cooperation. Therefore, the next inflection point of carbon productivity increase has not yet appeared.

Third, economic growth has a threshold effect between DSS and carbon productivity. With the economic growth of manufacturing, DSS has a positive "U" feature of first inhibiting and then promoting carbon productivity. In the early years of the manufacturing economy, most of its industrial structure was resource-intensive or labor-intensive, which increased energy consumption [27,72]. Thus, DSS had an inhibitory impact on carbon productivity at this stage. With the economic growth of manufacturing, its industrial structure has become technology-intensive, and DSS has entered a stage of rapid development. It has improved energy efficiency and carbon productivity. As the manufacturing economy

grows rapidly beyond the upper limit of its own production elasticity, carbon emissions increase, and carbon productivity decreases.

Fourthly, through the mediation effect test, it was found that technological innovation and industrial structure have partial mediation effects between DSS and carbon productivity. Therefore, manufacturing can increase its investment in technological R&D and optimize its industrial structure, thereby increasing the contribution of DSS to carbon productivity.

## 6. Conclusions and Policy Recommendations

### 6.1. Conclusions

This paper identifies the relationship between DSS and carbon productivity. In contrast to earlier studies of carbon productivity, this paper is not limited to the effects of single factors of digitalization or servitization. It is based on synergy theory and explores the role of the synergistic effect of both digitalization and servitization factors on carbon productivity. The empirical test concludes that there is a non-linear relationship between DSS and carbon productivity [19,73–75]. DSS has a positive "U" relationship with carbon productivity when economic growth is the threshold effect. The conclusion is similar to the research of other scholars [73,76]. Based on the contribution of digitalization and servitization to carbon productivity [8,9] and the development mechanism of digitalization and servitization [77,78], this paper further explores the effect of DSS on carbon productivity. The conclusion contributes to the carbon productivity development pathway. DSS can be used to improve carbon productivity, and it is an implementable path that can promote economic growth and environmental protection together. It provides a strong impetus for the sustainable development of manufacturing and a theoretical basis for global sustainable development.

This paper considers the influence path of DSS to enhance carbon productivity from multiple perspectives. Unlike previous studies considering a single variable, this paper considers both technological innovation and industrial structure to verify their mediating role in DSS on carbon productivity [15,36–38]. Based on the environmental Kuznets curve theory, the threshold effect is further explored [73,79]. Therefore, it can provide multiple implementable paths for DSS to enhance carbon productivity by considering various aspects such as technological innovation, industrial structure, and economic growth. It provides feasible solutions for manufacturing to achieve carbon peaking and carbon neutrality.

This paper provides valuable suggestions for the sustainable development of manufacturing. Firstly, manufacturing can develop DSS according to the actual situation of the enterprise. Instead of focusing on the development of one side only, managers use the dynamic development of digitalization and servitization to drive the development of enterprise DSS [80]. Heavily polluting enterprises can complete their transformation and development as quickly as possible through DSS. Enterprises can deepen the use of technology, knowledge, and other resources through digitalization and servitization collaboration platforms to maximize the use of digitalization and servitization and solve the problem of economic and environmental sustainability in manufacturing. Second, the research results verify the threshold effect of economic growth. Managers should always pay attention to the upper limit of production elasticity when carrying out DSS development [73]. It is true that DSS can enhance carbon productivity, but if enterprises pursue economic growth too quickly, it will be difficult for them to achieve sustainable development.

### 6.2. Policy Recommendations

The above conclusions provide a feasible path for manufacturing to enhance carbon productivity and develop a green economy. This paper examines the implementation path for DSS to enhance carbon productivity. This provides a sustainable development path for manufacturing, including low-carbon development enterprises. It also provides policymakers with a strategy for economically and environmentally sustainable development. The implementation path to boost carbon productivity through the development of DSS is as follows.

First, manufacturing DSS is developed, and the level of DSS is improved. At present, synergies are low in most provinces. Therefore, the government, through policy guidance, increases the support for the development of DSS to attract more enterprises to participate in the development of DSS. At the same time, it strengthens the publicity and promotion of DSS development, enhances the awareness and participation of enterprises, and promotes DSS. Manufacturing actively establishes a flexible organizational structure and accelerates the construction of advanced production factors, such as human resources and technology, through resource integration, information sharing, and technology research and development. DSS development conditions are then constructed to accelerate DSS development in manufacturing.

Second, DSS should be made the most out of, and the carbon productivity of manufacturing should be promoted with DSS. Simultaneously with economic development, manufacturing should rationally develop DSS according to its own economic development level. Along with rapid economic development, manufacturing should be promoted to shift to technology-intensive industries, changing the economic development model. The DSS level of manufacturing should be accelerated, improving energy utilization efficiency and, thus, carbon productivity.

Thirdly, optimizing industrial structure and improving technological innovation are important links for DSS in manufacturing to promote carbon productivity. The government should support the collaborative development of technological innovation capabilities carried out by enterprises and research institutions and promote the development of manufacturing in the direction of low energy consumption and low pollution and high efficiency. Manufacturing should adhere to a combination of market regulation and policy guidance, make full use of the basic role of the market in allocating resources, and achieve the optimal allocation of resources. Policy guidance should be used to promote cooperation and exchanges between all links and the outside world, increase investments in research and development, and promote the sustainable development of manufacturing.

Fourth, the East and the Midwest should carry out overall planning, reasonably guide the inflow of resources, reduce regional differences, and improve the DSS level of China's manufacturing. The government should exert its role in the regulation and management of resources to prevent a large amount of resources from flowing into developed provinces, which would further aggravate regional differences. The government of the Midwest can increase investment incentives, guide the inflow of resources between regions, and avoid the surplus of resources in developed regions. Cooperation and alliances between enterprises in various regions are encouraged, the maximization of redundant resources should be promoted, and the DSS level of manufacturing in each region should be improved.

*6.3. Limitation and Future Research*

This study starts from the synergy of digitalization and servitization to derive a feasible path to enhance carbon productivity. However, the synergistic path of digitalization and servitization has not been explored in depth in the synergistic development of digitalization and servitization. The characteristics of digitalization and servitization in the synergistic path with dynamic development can be further explored in subsequent related studies. In addition, the impact of corporate greening on carbon productivity has been studied. We can further explore the synergy between digitalization and greening based on the synergy theory. Hence, these matters warrant further discussion and exploration in future research endeavors.

**Author Contributions:** Conceptualization, G.L. and Y.C. (Yanan Chen); methodology, Y.C. (Yanan Chen); software, Y.C. (Yanan Chen); validation, G.L. and Y.C. (Yanan Chen); formal analysis, G.L.; investigation, G.L.; resources, G.L.; data curation, Y.C. (Yanan Chen); writing—original draft preparation, G.L., Y.C. (Yanan Chen) and Y.C. (Yan Cheng); writing—review and editing, G.L. and Y.C. (Yanan Chen); visualization, Y.C. (Yanan Chen); supervision, G.L.; project administration, G.L.; funding acquisition, G.L. All authors have read and agreed to the published version of the manuscript.

**Funding:** This work was supported by the "Shaanxi Social Science Fund" of China, grant number 2023SJ01.

**Data Availability Statement:** The data presented in this study are available upon request from the corresponding author.

**Conflicts of Interest:** The authors declare no conflicts of interest.

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
