# Peer review of "Can the Synergy of Digitalization and Servitization Boost Carbon-Related Manufacturing Productivity? Evidence from China’s Provincial Panel Data"

_sustainability, doi:10.3390/su16072655_

Round 1

Reviewer 1 Report

Comments and Suggestions for Authors

Dear Colleagues, the article has scientific and practical innovation. The article would be greatly useful if the authors:

1. write the scope of application of the presented results

2. . in conclusion, write further directions for research in continuation of this topic

3. the introduction does not explain in detail what literature review the presented hypotheses are based on

The topic of the article is relevant, however, the material itself requires improvement. It is important to build the structure of the abstract according to the structure of the article. The comma is indicated. The research methods are comma, the results are drawn. And the conclusions drawn are quite narrow. The presented material in the Results section requires facts and analytics within the framework of the presented topic. Conclusion It is important to conduct a literature review using Scopus articles and reflect. What are the works of leading scientific scientists on the topic of the article. Conclusion is important to indicate the purpose, scientific AND practical significance. In conclusion, it is necessary to draw conclusions on the theoretical and practical part.

Author Response

Thank you for giving us the opportunity to make revisions.

Point 1:

Write the scope of application of the presented results.

Response 1:

Thank you for your suggestion. We all think your comments are very reasonable. We have increased the scope of application of articles in section 6.1. We draw on the research of other scholars, based on their research direction, do in-depth research, broaden the research ideas.The modifications are made as follows:

 "The paper identifies the relationship between DSS and carbon productivity. In contrast to earlier studies of carbon productivity, the paper is not limited to the effect of single factors of digitalization or servitization on it. It is based on the synergy theory and explores the role of the synergistic effect of both digitalization and servitization factors on carbon productivity. The empirical test concludes that there is a non-linear relationship between DSS and carbon productivity. DSS has a positive “U” relationship with carbon productivity when economic growth is the threshold effect. The conclusion is similar to the research of other scholars. Based on the contribution of digitalization and servitization to carbon productivity and the development mechanism of digitalization and servitization, the paper further explores the path of DSS on carbon productivity. The conclusion contributes to the carbon productivity development pathway. DSS to improve carbon productivity as an implementable path to promote economic growth and environmental protection together. It provides a strong impetus for the sustainable development of the manufacturing and a theoretical basis for global sustainable development.

The paper considers the influence path of DSS to enhance carbon productivity from multiple perspectives. Unlike in previous studies considering a single variable, the paper considers both technological innovation and industrial structure t to verify their mediating role in DSS on carbon productivity. Based on the environmental Kuznets curve theory, the threshold effect is further explored. Therefore, it can provide multiple implementable paths for DSS to enhance carbon productivity by considering various aspects such as technological innovation, industrial structure and economic growth. It provides feasible solutions for the manufacturing to achieve carbon peaking and carbon neutrality.

The paper provides valuable suggestions for the sustainable development of the manufacturing. Firstly, the manufacturing can develop DSS according to the actual situation of the enterprise. Instead of focusing on the development of one side only, managers use the dynamic development of digitalization and servitization to drive the development of enterprise DSS. Heavily polluting enterprises can complete their transformation and development as quickly as possible through DSS. Enterprises can deepen the use of technology, knowledge and other resources through digitalization and servitization collaboration platforms to maximize the use of digitalization and servitization drive. Solve the problem of economic and environmental sustainability in manufacturing. Second, the research results verify the threshold effect of economic growth. Managers should always pay attention to their upper limit of production elasticity when conducting DSS development. It is true that DSS can enhance carbon productivity, but if enterprises pursue economic growth too quickly, it will be difficult for them to achieve sustainable development."

Point 2:

In conclusion, write further directions for research in continuation of this topic.

Response 2:

Thank you for your suggestion. We have modified the conclusion section by adding 6.3 Future Directions to supplement the article with future researchable directions for subsequent researchers to make theoretical references. The modifications are made as follows:

"The study starts from the synergy of digitalization and servitization to derive a feasible path to enhance carbon productivity. However, the synergistic path of digitalization and servitization has not been explored in depth in the synergistic development of digitalization and servitization. The characteristics of digitalization and servitization in the synergistic path with dynamic development can be further explored in subsequent related studies. In addition, the impact of corporate greening on carbon productivity has been studied. We can further explore the synergy between digitalization and greening based on the synergy theory. Hence, these matters warrant further discussion and exploration in future research endeavors."

Point 3:

The introduction does not explain in detail what literature review the presented hypotheses are based on. 

Response 3:

Thank you for your suggestion. We have revised the introductory section and added relevant literature for further clarification. We explain in detail what literature review the presented hypotheses are based on. The modifications are placed in paragraph 4 of Part I. The modifications are made as follows:

"Most of the current research involving carbon productivity focuses on the enhancing effect of digitalization. Digitalization enhances carbon productivity by promoting technological innovation and industrial structure. There are fewer studies on the enhancing effect of servitization on carbon productivity. The research on the specific path of enhancing carbon productivity is unclear. Current research mostly focuses on a single dimension of digitalization or servitisation to verify the impact on carbon productivity, and there is a lack of relevant research on both dimensions. "

Point 4:

The topic of the article is relevant, however, the material itself requires improvement.

Response 4:

Thank you for your suggestion. In order to better fit the topic, we have modified the introduction section of the article by adding references to further detail the relevance of this article to the topic. The modifications are made as follows:

"Serious environmental degradation has accompanied economic development. Economic growth has increased the emission of greenhouse gases, of which CO2 is a representative, exacerbating the harm to the ecosystem. Environmental protection has also become a widespread concern in the world. China has become a large CO2 emitting country due to its rapid economic development and industrialization. China proposes carbon peaking and carbon neutrality targets to achieve global environmental protection. In order to steadily achieve the goal, China manufacturing needs to carry out green and sustainable development. Carbon productivity improvement can be used as a sustainable development indicator to coordinate economic growth and reduce CO2 emissions. It is a sustainable development path for the green development of China manufacturing."

Cite other relevant references to further elaborate the article's relevance to the topic.In the conclusion section, more references are cited to support the conclusions of this article to ensure its scientific validity. We added no less than 20 references to increase the relevance of the articles.

Point 5:

It is important to build the structure of the abstract according to the structure of the article. The comma is indicated. The research methods are comma, the results are drawn. 

Response 5:

Thank you for your suggestion. We modified the abstract section according to the structure of the article, before presenting the research methodology and drawing conclusions.We have revised and supplemented the study methods and the conclusions drawn.The modifications are made as follows:

"Based on the data of manufacturing in 30 provinces in China from 2013 to 2020, the coupled coordination degree model is used to calculate the degree of coordination of manufacturing. A regression effects model is used to explore the intrinsic mechanism of the impact of the synergy of digitalization and servitization on carbon productivity. "

Point 6:

The presented material in the Results section requires facts and analytics within the framework of the presented topic.

Response 6:

Thank you for your suggestion. We have further analysed the article based on its results and added relevant references for support.

We explain each empirical result in Chapter 5. The possible reasons for this result are responded to and references similar to this empirical result are cited. We provide further analysis based on the empirical results in section 6.1 to further analyse the research implications and future directions of this paper.

Point 7:

It is necessary to draw conclusions on the theoretical and practical part.

Response7:

Thank you for your suggestion. We have modified the section. We have modified the conclusion section by adding two sections on conclusion, scope of application and future directions. We added section 6.1 as a summary of the article, presenting the research significance and innovations of the paper.

The theoretical part:"The paper identifies the relationship between DSS and carbon productivity. In contrast to earlier studies of carbon productivity, the paper is not limited to the effect of single factors of digitalization or servitization on it. It is based on the synergy theory and explores the role of the synergistic effect of both digitalization and servitization factors on carbon productivity. The empirical test concludes that there is a non-linear relationship between DSS and carbon productivity. DSS has a positive “U” relationship with carbon productivity when economic growth is the threshold effect. The conclusion is similar to the research of other scholars. Based on the contribution of digitalization and servitization to carbon productivity and the development mechanism of digitalization and servitization, the paper further explores the path of DSS on carbon productivity. The conclusion contributes to the carbon productivity development pathway. DSS to improve carbon productivity as an implementable path to promote economic growth and environmental protection together. It provides a strong impetus for the sustainable development of the manufacturing and a theoretical basis for global sustainable development.

The paper considers the influence path of DSS to enhance carbon productivity from multiple perspectives. Unlike in previous studies considering a single variable, the paper considers both technological innovation and industrial structure t to verify their mediating role in DSS on carbon productivity. Based on the environmental Kuznets curve theory, the threshold effect is further explored. Therefore, it can provide multiple implementable paths for DSS to enhance carbon productivity by considering various aspects such as technological innovation, industrial structure and economic growth. It provides feasible solutions for the manufacturing to achieve carbon peaking and carbon neutrality. "

The practical part:"The paper provides valuable suggestions for the sustainable development of the manufacturing. Firstly, the manufacturing can develop DSS according to the actual situation of the enterprise. Instead of focusing on the development of one side only, managers use the dynamic development of digitalization and servitization to drive the development of enterprise DSS. Heavily polluting enterprises can complete their transformation and development as quickly as possible through DSS. Enterprises can deepen the use of technology, knowledge and other resources through digitalization and servitization collaboration platforms to maximize the use of digitalization and servitization drive. Solve the problem of economic and environmental sustainability in manufacturing. Second, the research results verify the threshold effect of economic growth. Managers should always pay attention to their upper limit of production elasticity when conducting DSS development. It is true that DSS can enhance carbon productivity, but if enterprises pursue economic growth too quickly, it will be difficult for them to achieve sustainable development."

Thank you again sincerely. If there are any imperfections in this manuscript, we hope you will help point them out. We will do our best to revise it.

Reviewer 2 Report

Comments and Suggestions for Authors

First, I must confess that I could not review a manuscript of this kind being unbiased, as I am a firm believer that the planet needs a zero (0) carbon use policy and the gradual removal of carbon. This is despite the fact that I am a process engineer.

I do not agree that this work is part of a contribution to the sustainability of the planet, as stated in the objectives and scope of the journal. In addition to the above, there is no doubt that the manuscript has a political focus, which I believe should be completely removed from an academic paper such as the one planned for publication.

With the above, I am not saying that the paper was poorly written or that it lacks quality; only that I consider that it is not pertinent for the journal.

In addition to the above, I leave some specific comments:

L18: so how is the relationship? In several sections of the manuscript, assertions like this are left unexplained.

Keywords: Words contained in the title should not be repeated. In this case, all the keywords are in the title, which is a waste of opportunity, since this would not help in any way to take advantage of the raison of keywords, which is to improve the appearance of the manuscript in search engines.

L35: omit, especially if not referenced.

. I recommend using significant figures in Tables and the rest of the text.

L483-484: This is a no-brainer. Avoid.

487: Same as L18.

. Much of the Conclusions is a repetition of the results, which is not the purpose of this section.

Author Response

Thank you for giving us the opportunity to make revisions.

Point 1:

There is no doubt that the manuscript has a political focus, which I believe should be completely removed from an academic paper such as the one planned for publication.

Response 1:

Thank you for pointing out this problem for us, and we have removed the political focus from our manuscript. I deleted something as follows:

"The report of the 20th National Congress of the Communist Party of China proposed to steadily promote the realization of carbon peak and carbon neutrality"

Point 2:

With the above, I am not saying that the paper was poorly written or that it lacks quality; only that I consider that it is not pertinent for the journal.

Response 2:

Thank you for your suggestion. Sorry, perhaps I didn't write in enough depth to make you think my article was not relevant to the topic. I have made changes to address the introductory section that describes how my article relates to the topic. Our addition is in the first paragraph of chapter 1, which reads as follows: "Serious environmental degradation has accompanied economic development. Economic growth has increased the emission of greenhouse gases, of which CO2 is a representative, exacerbating the harm to the ecosystem. Environmental protection has also become a widespread concern in the world. China has become a large CO2 emitting country due to its rapid economic development and industrialization. China proposes carbon peaking and carbon neutrality targets to achieve global environmental protection. In order to steadily achieve the goal, China manufacturing needs to carry out green and sustainable development. Carbon productivity improvement can be used as a sustainable development indicator to coordinate economic growth and reduce CO2 emissions. It is a sustainable development path for the green development of China manufacturing. "

The manufacturing industry has shifted from resource-intensive to technology-intensive industries by optimising its industrial structure and reducing its dependence on energy. This reduced energy consumption promotes increased carbon productivity in the manufacturing sector. Reduce carbon emissions from China's manufacturing companies and achieve carbon neutrality goals at an early date. Help China's manufacturing industry to take the path of sustainable development.

Point 3:

L18:so how is the relationship? In several sections of the manuscript, assertions like this are left unexplained.

Response 3:

Thank you for your suggestion. We preliminarily verified the non-linear relationship between the DSS and carbon productivity by regression effect model in part 4.3.3, which verified the conjecture of H1. In part 4.3.7, by verifying the threshold effect of economic growth, it is concluded that the DSS has a positive U relationship with carbon productivity under the threshold benefit of economic growth, and H3 is verified. Therefore, for the relationship between the DSS and carbon productivity we argue in the manuscript that there is a non-linear relationship. Under the threshold effect of economic growth, the two have a positive U relationship.

Point 4:

 Keywords: Words contained in the title should not be repeated.

Response 4:

Thank you for your suggestion. We have modified the keywords according to the core content of the article. We have changed "digitalization; servitization" to "coupling coordination degree; positive ‘U’ relationships".

Point 5:

 L35: omit, especially if not referenced.

Response 5:

Thank you for your suggestion. We did miss the citation for this data. We have partially revised based on the content of the article. Sentences that need to be quoted are all supplemented with references to ensure the accuracy of each sentence. The modifications are made as follows:

"Serious environmental degradation has accompanied economic development[1]. Economic growth has increased the emission of greenhouse gases[2], of which CO2 is a representative, exacerbating the harm to the ecosystem. Environmental protection has also become a widespread concern in the world. China has become a large CO2 emitting country due to its rapid economic development and industrialization[3]. China proposes carbon peaking and carbon neutrality targets to achieve global environmental protection[4]. In order to steadily achieve the goal, China manufacturing needs to carry out green and sustainable development[5]. Carbon productivity improvement can be used as a sustainable development indicator to coordinate economic growth and reduce CO2 emissions. It is a sustainable development path for the green development of China manufacturing[6]." 

Point 6:

I recommend using significant figures in Tables and the rest of the text.

Response 6:

Thank you for your suggestion. We using significant figures for the DSS data(Table 6). The digitalization data (Table 3) and the servitization data (Table 5) are the sources of the data in Table 6. Because the review 3 teacher believes that digitalization and servitization are important data sources for the DSS, so we did not use significant figures in tables 3 and 5. The modifications are made as follows:

Point 7:

L483-484: This is a no-brainer. Avoid.

L487: Same as L18.

Response 7:

Thank you for your suggestion. We have modified the section to avoid this type of error and increase the readability of our manuscripts.

Point 8:

Much of the Conclusions is a repetition of the results, which is not the purpose of this section. 

Response 8:

Thank you for your suggestion. We have revised the section. We modified the conclusions section by adding two additional sections on conclusions and future directions. We have modified the original conclusions as a separate section for the discussion of empirical results as Chapter 5. Conclusions and future directions and policy recommendations were added as separate chapters 6. We have added no less than 20 references to support this conclusion. The modifications are made as follows:

Discussions

The paper empirically analyzes the influence of manufacturing DSS on carbon productivity in 30 provinces of China from 2013 to 2020 and divides the manufacturing DSS levels. The conclusion is as follows.

First, the level of DSS in China's manufacturing is low, and the average level is still far from the high level of cooperation. China's manufacturing DSS faces the challenge of uneven development, with obvious regional differences. In developed eastern regions like Beijing, Guangdong, and and other regions, the level of DSS implementation surpasses that in less developed areas, yet it still has a certain gap from the high level of coordination.

Second, DSS makes a notable positive contribution to carbon productivity, and the two show a strong and then weak non-linear relationship. The reason is that the development of DSS relies on electricity, computing power, high and new technologies and others. These supporting factors require a large amount of energy consumption to a certain extent, and the technology development cycle is long, which will hinder the contribution of DSS to carbon productivity in the short term. At present, DSS in most provinces is at a low level, and DSS in developed provinces such as Beijing and Guangdong is only at a general level, and has not reached a high level of cooperation. Therefore, the next inflection point of carbon productivity increase has not yet appeared.

Third, economic growth has a threshold effect between DSS and carbon productivity. With the economic growth of the manufacturing, DSS has a positive “U” feature of first inhibiting and then promoting carbon productivity. In the early years of the manufacturing economy, most of its industrial structure is resource-intensive or labor-intensive, which increases energy consumption. So DSS has an inhibitory impact on carbon productivity at this stage. With the economic growth of the manufacturing, its industrial structure has become technology-intensive, and DSS has entered a stage of rapid development. It has improved energy efficiency and carbon productivity. As the manufacturing economy grows rapidly beyond the upper limit of its own production elasticity, carbon emissions increase and carbon productivity decreases.

Fourthly, through the mediation effect test, it is found that technological innovation and industrial structure have partial mediation effect between DSS and carbon productivity. Therefore, the manufacturing can increase its investment in technological R&D and optimize its industrial structure, thereby increasing the contribution of DSS to carbon productivity.

Conclusions

The paper identifies the relationship between DSS and carbon productivity. In contrast to earlier studies of carbon productivity, the paper is not limited to the effect of single factors of digitalization or servitization on it. It is based on the synergy theory and explores the role of the synergistic effect of both digitalization and servitization factors on carbon productivity. The empirical test concludes that there is a non-linear relationship between DSS and carbon productivity. DSS has a positive “U” relationship with carbon productivity when economic growth is the threshold effect. The conclusion is similar to the research of other scholars. Based on the contribution of digitalization and servitization to carbon productivity and the development mechanism of digitalization and servitization, the paper further explores the path of DSS on carbon productivity. The conclusion contributes to the carbon productivity development pathway. DSS to improve carbon productivity as an implementable path to promote economic growth and environmental protection together. It provides a strong impetus for the sustainable development of the manufacturing and a theoretical basis for global sustainable development.

The paper considers the influence path of DSS to enhance carbon productivity from multiple perspectives. Unlike in previous studies considering a single variable, the paper considers both technological innovation and industrial structure t to verify their mediating role in DSS on carbon productivity. Based on the environmental Kuznets curve theory, the threshold effect is further explored. Therefore, it can provide multiple implementable paths for DSS to enhance carbon productivity by considering various aspects such as technological innovation, industrial structure and economic growth. It provides feasible solutions for the manufacturing to achieve carbon peaking and carbon neutrality.

The paper provides valuable suggestions for the sustainable development of the manufacturing. Firstly, the manufacturing can develop DSS according to the actual situation of the enterprise. Instead of focusing on the development of one side only, managers use the dynamic development of digitalization and servitization to drive the development of enterprise DSS. Heavily polluting enterprises can complete their transformation and development as quickly as possible through DSS. Enterprises can deepen the use of technology, knowledge and other resources through digitalization and servitization collaboration platforms to maximize the use of digitalization and servitization drive. Solve the problem of economic and environmental sustainability in manufacturing. Second, the research results verify the threshold effect of economic growth. Managers should always pay attention to their upper limit of production elasticity when conducting DSS development. It is true that DSS can enhance carbon productivity, but if enterprises pursue economic growth too quickly, it will be difficult for them to achieve sustainable development.

Limitation and future research

The study starts from the synergy of digitalization and servitization to derive a feasible path to enhance carbon productivity. However, the synergistic path of digitalization and servitization has not been explored in depth in the synergistic development of digitalization and servitization. The characteristics of digitalization and servitization in the synergistic path with dynamic development can be further explored in subsequent related studies. In addition, the impact of corporate greening on carbon productivity has been studied. We can further explore the synergy between digitalization and greening based on the synergy theory. Hence, these matters warrant further discussion and exploration in future research endeavors." 

Thank you again for reading our manuscript very carefully and pointing out our shortcomings, which helped us a lot to improve the quality of our manuscript.

Reviewer 3 Report

Comments and Suggestions for Authors

Abstract is too long. The reader loses focus of the research topic or title. The essence of the Abstract is the setting of the hypothesis, pointing the chosen methodology, the conducted research, the simulation or calculation and the presentation of the results with the final conclusion.

Unfortunately, the manuscript is full of errors such as joining words with square brackets of reference literature. This should be separated because it interferes with reading and concentration.

Although this scientific research is actually well structurally described and set up, it relies on the research and coefficients of other scientists (ref.lit.44, 45, 51, 52, ...) and the data source from the period 2013-2020.

Information about the references of equations (2), (3) and (4) is missing. Authors should add the origin source of those equations.

Too many errors in the text from line 295 to line 303.

Table 4 is the most important data source in this research study.

Finally: this paper empirically analyzes the impact of DSS production on carbon productivity in 30 provinces in China from 2013 to 2020. It would be more correct from the perspective of respecting the construction of the scientific work to designate the current Chapter No.5. as deliberation/consideration and target incentives/or national policy obligations, and finally Chapter No.6. as a conclusion that the authors autonomously deliver the final word in two short sentences.

Comments on the Quality of English Language

Moderate English language editing required. Quite a few sentences that are too long should be paraphrased to make them more understandable.

Author Response

Thank you for giving us the opportunity to make revisions.

Point 1:

Abstract is too long. The reader loses focus of the research topic or title.

Response 1:

Thank you for your suggestion. We agree with your opinion, so we have revised the abstract section by deleting or shortening some of the text and shortening the length of the abstract. The modifications are made as follows:

"Under the goal of carbon peaking and carbon neutrality, carbon productivity has become a means of sustainability in the manufacturing, and the impact of the synergy of digitalization and servitization(DSS) on carbon productivity(CP) deserves in-depth study. Based on the data of manufacturing in 30 provinces in China from 2013 to 2020, the coupled coordination degree model is used to calculate the degree of coordination of manufacturing. A regression effects model is used to explore the intrinsic mechanism of the impact of DSS on CP. The main results show: (1) The DSS in manufacturing positively contributes to enhancing CP, and there are non-linear features in both. (2) Technological innovation can contribute to the impact of DSS on CP, as does industry structure, and there is a mediating effect between the two. (3) When economic growth is used as the threshold, DSS and CP reflect a positive “U” relationship. Based on the above findings, policy recommendations are made to promote sustainable development of the manufacturing."

Point 2:

The manuscript is full of errors such as joining words with square brackets of reference literature.

Response 2:

Thank you for pointing out this problem for us, and we have followed the MDPI reference format for the references in this paper.

E.g., Ma R.,Lin B. Digitalization and energy-saving and emission reduction in Chinese cities: Synergy between industrialization and digitalization. Applied Energy. 2023.345,121308.https://doi.org/10.1016/j.apenergy.2023.121308.

Point 3:

Although this scientific research is actually well structurally described and set up, it relies on the research and coefficients of other scientists (ref.lit.44, 45, 51, 52, ...) and the data source from the period 2013-2020.

Response 3:

Thank you for your suggestion. We have made additions to the data sources for the DSS (Table 6) for the sake of data completeness(Table 3 and Table 5). The data in Table 6 is calculated based on digitalization and servitization through the coupling degree formula. The source of the raw data for all data is described in section 3.2.3 to ensure the authenticity of the data.

Point 4:

Information about the references of equations (2), (3) and (4) is missing. Authors should add the origin source of those equations.

Response 4:

Thank you for your suggestion. We have added formulas for which sources are lacking. Equations (2) and (3) together with (1) are the overall formulas for the coupled coordination degree. We cite the source of formulae in the introductory section of equation (1). We have supplemented the source of equation (4) by citing relevant references.

Point 5:

Too many errors in the text from line 295 to line 303.

Response 5:

Thank you for your suggestion. The errors in this section were corrected and the overall article was checked.

Point 6:

Table 4 is the most important data source in this research study.

Response 6:

Thank you for your suggestion. The data in Table 4 were derived using the digitalization and servitization data scores through the coupled coordination degrees. The digitalization and servitization score data were not placed within the main text due to space considerations. To ensure data integrity, we have refined this by placing the digitalization(Table 3) and servitization(Table 5) score data into sections 3.2.1 and 3.2.2 respectively.

Point 7:

It would be more correct from the perspective of respecting the construction of the scientific work to designate the current Chapter No.5. as deliberation/consideration and target incentives/or national policy obligations, and finally Chapter No.6. as a conclusion that the authors autonomously deliver the final word in two short sentences.

Response 7:

Thank you for your suggestion. We have modified the conclusions section by adding two additional sections on conclusions and future directions. The original conclusions were modified to include a discussion of the empirical results as a separate section as Chapter 5. The new conclusions and future directions and policy recommendations are added as a separate chapter 6. We have added no less than 20 references to support this conclusion. The modifications are made as follows:

"5. Discussions

The paper empirically analyzes the influence of manufacturing DSS on carbon productivity in 30 provinces of China from 2013 to 2020 and divides the manufacturing DSS levels. The conclusion is as follows.

First, the level of DSS in China's manufacturing is low, and the average level is still far from the high level of cooperation. China's manufacturing DSS faces the challenge of uneven development, with obvious regional differences. In developed eastern regions like Beijing, Guangdong, and and other regions, the level of DSS implementation surpasses that in less developed areas, yet it still has a certain gap from the high level of coordination.

Second, DSS makes a notable positive contribution to carbon productivity, and the two show a strong and then weak non-linear relationship. The reason is that the development of DSS relies on electricity, computing power, high and new technologies and others. These supporting factors require a large amount of energy consumption to a certain extent, and the technology development cycle is long, which will hinder the contribution of DSS to carbon productivity in the short term. At present, DSS in most provinces is at a low level, and DSS in developed provinces such as Beijing and Guangdong is only at a general level, and has not reached a high level of cooperation. Therefore, the next inflection point of carbon productivity increase has not yet appeared.

Third, economic growth has a threshold effect between DSS and carbon productivity. With the economic growth of the manufacturing, DSS has a positive “U” feature of first inhibiting and then promoting carbon productivity. In the early years of the manufacturing economy, most of its industrial structure is resource-intensive or labor-intensive, which increases energy consumption. So DSS has an inhibitory impact on carbon productivity at this stage. With the economic growth of the manufacturing, its industrial structure has become technology-intensive, and DSS has entered a stage of rapid development. It has improved energy efficiency and carbon productivity. As the manufacturing economy grows rapidly beyond the upper limit of its own production elasticity, carbon emissions increase and carbon productivity decreases.

Fourthly, through the mediation effect test, it is found that technological innovation and industrial structure have partial mediation effect between DSS and carbon productivity. Therefore, the manufacturing can increase its investment in technological R&D and optimize its industrial structure, thereby increasing the contribution of DSS to carbon productivity.

6.Conclusions and Policy Recommendations

6.1. Conclusions

The paper identifies the relationship between DSS and carbon productivity. In contrast to earlier studies of carbon productivity, the paper is not limited to the effect of single factors of digitalization or servitization on it. It is based on the synergy theory and explores the role of the synergistic effect of both digitalization and servitization factors on carbon productivity. The empirical test concludes that there is a non-linear relationship between DSS and carbon productivity. DSS has a positive “U” relationship with carbon productivity when economic growth is the threshold effect. The conclusion is similar to the research of other scholars. Based on the contribution of digitalization and servitization to carbon productivity and the development mechanism of digitalization and servitization, the paper further explores the path of DSS on carbon productivity. The conclusion contributes to the carbon productivity development pathway. DSS to improve carbon productivity as an implementable path to promote economic growth and environmental protection together. It provides a strong impetus for the sustainable development of the manufacturing and a theoretical basis for global sustainable development.

The paper considers the influence path of DSS to enhance carbon productivity from multiple perspectives. Unlike in previous studies considering a single variable, the paper considers both technological innovation and industrial structure t to verify their mediating role in DSS on carbon productivity. Based on the environmental Kuznets curve theory, the threshold effect is further explored. Therefore, it can provide multiple implementable paths for DSS to enhance carbon productivity by considering various aspects such as technological innovation, industrial structure and economic growth. It provides feasible solutions for the manufacturing to achieve carbon peaking and carbon neutrality.

The paper provides valuable suggestions for the sustainable development of the manufacturing. Firstly, the manufacturing can develop DSS according to the actual situation of the enterprise. Instead of focusing on the development of one side only, managers use the dynamic development of digitalization and servitization to drive the development of enterprise DSS. Heavily polluting enterprises can complete their transformation and development as quickly as possible through DSS. Enterprises can deepen the use of technology, knowledge and other resources through digitalization and servitization collaboration platforms to maximize the use of digitalization and servitization drive. Solve the problem of economic and environmental sustainability in manufacturing. Second, the research results verify the threshold effect of economic growth. Managers should always pay attention to their upper limit of production elasticity when conducting DSS development. It is true that DSS can enhance carbon productivity, but if enterprises pursue economic growth too quickly, it will be difficult for them to achieve sustainable development.

6.2. Policy Recommendations

The above conclusions provide a feasible path for the manufacturing to enhance carbon productivity and develop the green economy. The paper examines the implementation path for DSS to enhance carbon productivity. This provides a sustainable development path for the manufacturing, including low carbon development enterprises. It also provides policy makers with a strategy for economically and environmentally sustainable development. The implementation path to boost carbon productivity through the development of DSS is as follows:

First, develop manufacturing DSS and improve the level of DSS. At present, synergies are low in most provinces. Therefore, the government, through policy guidance, increases the support for the development of DSS to attract more enterprises to participate in the development of DSS. At the same time, it strengthens the publicity and promotion of DSS development, enhances the awareness and participation of enterprises, and promotes DSS. The manufacturing actively establishes a flexible organizational structure and accelerates the construction of advanced production factors such as human resources and technology through resource integration, information sharing and technology research and development. Construct DSS development conditions to accelerate DSS development in manufacturing.

Second, making the most of DSS and promoting the carbon productivity of manufacturing with DSS. At the same time of economic development, the manufacturing should rationally develop DSS according to its own economic development level. Along with rapid economic development, the manufacturing should be promoted to shift to technology-intensive industries, change the economic development model. Accelerate DSS level of the manufacturing, improve energy utilization efficiency, and thus improve carbon productivity.

Thirdly, optimizing industrial structure and improving technological innovation are important links for DSS in manufacturing to promote carbon productivity. The Government should support the collaborative development of technological innovation capabilities by enterprises and research institutions and promote the development of the manufacturing in the direction of low energy and pollution consumption and high efficiency. The manufacturing should adhere to the combination of market regulation and policy guidance, make full use of the basic role of the market in allocating resources, and achieve optimal allocation of resources. Use policy guidance to promote cooperation and exchanges between all links and the outside world, increase investment in research and development and promote the sustainable development of the manufacturing.

Fourth, the East and the Midwest make overall planning, reasonably guide the inflow of resources, reduce regional differences, and improve DSS level of China’s manufacturing. The government should exert its role in the regulation and management of resources to avoid a large amount of resources flowing into developed provinces and further aggravate regional differences. The government of the Midwest can increase investment incentives, guide the inflow of resources between regions, and avoid the surplus of resources in the developed regions. Encourage cooperation and alliances between enterprises in various regions, promote the maximisation of redundant resources, and improve DSS level of manufacturing in each region.

6.3. Limitation and future research

The study starts from the synergy of digitalization and servitization to derive a feasible path to enhance carbon productivity. However, the synergistic path of digitalization and servitization has not been explored in depth in the synergistic development of digitalization and servitization. The characteristics of digitalization and servitization in the synergistic path with dynamic development can be further explored in subsequent related studies. In addition, the impact of corporate greening on carbon productivity has been studied. We can further explore the synergy between digitalization and greening based on the synergy theory. Hence, these matters warrant further discussion and exploration in future research endeavors."

Point 8:

Comments on the Quality of English Language.Moderate English language editing required. Quite a few sentences that are too long should be paraphrased to make them more understandable. 

Response 8:

Thank you for your suggestion. We revised the English of the article and sought outside help so that the article could be better understood.

Thank you again for all your suggestions for us, we think your suggestions have helped us to improve the quality of our manuscript. If there is still a shortage somewhere, we hope you can point it out again and we will do our best to improve it.

Round 2

Reviewer 2 Report

Comments and Suggestions for Authors

The authors provide responses to comments.